# Decontamination of N95 and surgical masks using a treatment based on a continuous gas phase-Advanced Oxidation Process

**Mahdiyeh Hasani[1], Tracey Campbell[2], Fan Wu[1], Keith Warriner[1]***

**1** Department of Food Science, University of Guelph, Guelph, Ontario, Canada, **2** Centre for Microbial Chemical Biology, McMaster University, Hamilton, Canada

* kwarrine@uoguelph.ca

**Data Availability Statement:** All relevant data are within the paper and its Supporting information files.

## Abstract

A gas-phase Advanced Oxidation Process (gAOP) was evaluated for decontaminating N95 and surgical masks. The continuous process was based on the generation of hydroxyl-radicals via the UV-C (254 nm) photo-degradation of hydrogen peroxide and ozone. The decontamination efficacy of the gAOP was dependent on the orientation of the N95 mask passing through the gAOP unit with those positioned horizontally enabling greater exposure to hydroxyl-radicals compared to when arranged vertically. The lethality of gAOP was independent of the applied hydrogen peroxide concentration (2–6% v/v) but was significantly ($P<0.05$) higher when $H_2O_2$ was introduced into the unit at 40 ml/min compared to 20 ml/min. A suitable treatment for N95 masks was identified as 3% v/v hydrogen peroxide delivered into the gAOP reactor at 40 ml/min with continuous introduction of ozone gas and a UV-C dose of 113 mJ/cm$^2$ (30 s processing time). The treatment supported >6 log CFU decrease in *Geobacillus stearothermophilus* endospores, > 8 log reduction of human coronavirus 229E, and no detection of *Escherichia coli* K12 on the interior and exterior of masks. There was no negative effect on the N95 mask fitting or particulate efficacy after 20 passes through the gAOP system. No visual changes or hydrogen peroxide residues were detected (<1 ppm) in gAOP treated masks. The optimized gAOP treatment could also support >6 log CFU reduction of endospores inoculated on the interior or exterior of surgical masks. *G. stearothermophilus* Apex spore strips could be applied as a biological indicator to verify the performance of gAOP treatment. Also, a chemical indicator based on the oxidative polymerization of pyrrole was found suitable for reporting the generation of hydroxyl-radicals. In conclusion, gAOP is a verifiable treatment that can be applied to decontaminate N95 and surgical masks without any negative effects on functionality.

## Introduction

In the CoVid-19 pandemic, caused by the spread of SARS-Cov-2, there was a shortage of personal protective equipment (PPE) such as N95 masks and surgical (procedure) masks [1–3]. The N95 mask consists of multiple filter layers that can filter 95% of particulates and provides

**Funding:** This work was supported by University of Guelph CoVid-19 Research and Catalyst Fund. The funder had no role in the study design, data collection, data analysis, decision to publish or preparation of the manuscript.

**Competing interests:** The authors have declared that no competing interests exist.

satisfactory respiratory protection against the virus [1,4]. Although surgical masks have a looser fit it has been reported that the filter layers also provide adequate protection against the virus within moderate-risk environments [4].

Given the shortage of N95 masks, the general use of surgical masks has been widely adopted for general use. Although surgical masks are intended for single use in moderate-risk environments there has been little attention on how to extend working life or recycle. This becomes relevant considering the demand for surgical masks coupled with the need to avoid non-compostable waste reaching landfills [5]. Consequently, there is a demand for mask decontamination methods that can be applied to enable recycling or extended use without an increased risk of contracting or disseminating infectious agents [6,7]. To this end, criteria have been developed to define the requirements of a suitable mask decontamination treatment that ensure adequate microbial inactivation while not negatively affecting functionality [8]. Specifically, there is a requirement that the process should support 6 log CFU reduction of *Geobacillus stearothermophilus* endospores and 4 log reduction of coronavirus or suitable surrogate. No chemical residues should be present on masks or change in functionality in fitting and particulate filtration.

There have been several successful N95 mask decontamination methods described based on microwave heating, ethylene oxide, hydrogen peroxide vapor, and Ultraviolet Germicidal irradiation (UVGI) [9–11]. Each method have advantages and disadvantages with regards to application, the effect on mask integrity, and antimicrobial efficacy [12]. For example, thermal treatments exhibit anti-viral activity although mask integrity can be compromised [13]. Similarly, UVGI has been shown to inactivate viruses such as H1N1 without compromising mask functionality [14]. However, the poor penetration of UV-C photons into the filter layers of the N95 masks requires high doses ($>270$ kJ/ m$^2$) to be applied that can result in 20–51% loss in strap strength [15,16]. Hydrogen peroxide vapor is the most widely used N95 mask decontamination method and has been demonstrated to support endospore reduction with negligible effects on mask functionality [17]. However, the process is a batch system that requires the application of $>480$ ppm hydrogen peroxide over an extended period resulting in a process that takes around 3 h.

A surgical mask decontamination method based on applying dry heat (60–70°C) for 1 has been reported [18]. Here, the researchers inoculated surgical masks with different bacteria (*Escherichia coli*, *Staphylococcus aureus*, *Pseudomonas aeruginosa*, *Klebsiella pneumonia*, *Acintobacter spp*, *Corybacterium spp*) yeast (*Candida albicans*) or virus (H1N1). The inoculation level per mask was 3 log CFU with the virus loading being estimated at 320 virons. No survivors were recovered from masks heated for 1 h at either 60 or 70°C although filtration efficacy was lost after 2–3 cycles [18]. However, the inactivation of bacterial endospores was not considered in the study.

A further study compared the decontamination efficacy of dry heat (90°C for 70 mins) and microwave generated steam to inactivate *S. aureus* on surgical masks [19]. Although both processes supported a $>6$ log reduction of *S. aureus* the microwave treatment resulted in a loss of surgical mask integrity after a single cycle. By using the dry heat method, the masks could tolerate 3 cycles without loss of function [19]. Again, the researchers did not determine efficacy against endospores given the thermal treatment was below that to support inactivation. In an alternative method, Quaternary ammonium salt surfactant has been applied to masks that supported a 2–3 log reduction of *S. aureus* although the health implication of chemical residues was not determined [20].

In the following, a process based on gas-phase Advanced Oxidation Process (gAOP) for N95 and surgical mask disinfection was evaluated. The gAOP process is derived from Fenton-like reactions that generate hydroxyl-radicals from the degradation of hydrogen peroxide and/

or ozone [21]. The Advanced Oxidation process has previously been applied for degrading pollutants in water treatment [22] with the gas-phase treatment (gAOP) receiving less attention. Yet, a gAOP process has a history of use in carton packaging decontamination whereby UV-C (254 nm) mediated degradation of 1% v/v hydrogen peroxide can support >5 log CFU inactivation of *Bacillus* endospores [23–25]. More recently, gAOP has been applied to inactivate pathogens such as *Listeria monocytogenes* on fresh fruit and vegetables [21,26]. Here, the fresh produce to be decontaminated is passed through a hydrogen peroxide spray into a reactor where ozone gas is continuously applied under the illumination of UV-C lamps. The key advantages of the gAOP process is the high anti-microbial efficacy with the ability to inactivate microbes on the surface and sub-surface without negatively affecting the item being sanitized [21,26]. In this respect, the gAOP process can be envisioned to combine the advantages of hydrogen peroxide vapor and UVGI in delivering a rapid method for decontaminating N95 masks without compromising functionality.

A further requirement of an N95 mask decontamination method is the need to verify treatment performance via a biological and/or chemical indicator. The former is satisfied with the inactivation of endospores such as *Bacillus pumilus* (in the case of irradiation-based treatments) or *G. stearothermophilus* (for thermal treatments). It has been reported that *B. pumilus and G. stearothermophilus* have comparable sensitivity to hydroxyl-radicals when applied in solution thereby suggesting that either would be suitable index organisms [23].

With regards to chemical indicators, the AOP process for water disinfection is based on actinometry such as methylene blue [27]. The main advantage of methylene blue is that it exhibits semi-selectivity towards hydroxyl-radicals but relatively insensitive to hydrogen peroxide or UV-C alone [27]. Yet as a chemical indicator, methylene blue is degraded to colorless via the AOP reaction when a positive color change would make it easier to identify decontaminated items. In the following study, pyrrole was evaluated as a chemical indicator to report on the gAOP treatment. Pyrrole can undergo oxidative polymerization to form a conducting polymer with the degree of conjugation being reflective by darkening [28]. In the course of the polymerization process an anionic dopant is incorporated into the film to counter the positive charge on the polypyrrole backbone. Nafion was used as the doping ion in the current study as the surfactant has been reported to support a homogenous polymerization process and adherent polypyrrole film [29].

It has been reported that hydroxyl-radicals, along with other strong oxidants such as persulfate, can support the oxidative polymerization of pyrrole thereby leading to an irreversible transition from colorless to brown-black [29]. Hence, it is conceivable that the oxidative polymerization of pyrrole can be applied to report on the gAOP decontamination process.

## Materials and methods

### Microbial cultivation and enumeration

*Escherichia coli* K12 was obtained from ATCC (Atlanta, US) and was stored at -80˚C in tryptic soy broth (TSB; Thermo Fisher, Whitby, Canada) containing 20% w/v glycerol. The bacteria were recovered by streaking out onto a tryptic soy agar (TSA; Thermo Fisher) plate that was incubated at 37˚C for up to 48 h. A colony was transferred to TSB that was incubated overnight at 37˚C and used to inoculate TSB that was incubated overnight. The cells were harvested by centrifugation (5000g for 10 min) and the cell pellet resuspended in saline to give a final cell density of 9 log CFU/ml ($OD_{600}$ 2.0 measured using a spectrophotometer; BioRad, Mississauga, Canada). The suspension of *E. coli* K12 was stored at 4˚C until required but used within a week.

*Geobacillus stearothermophilus* ATCC 7963 was cultivated on a TSA slant that was incubated at 55°C for 72 h. The growth was scraped off the agar layer and transferred to 10 ml sterilized distilled water and used to inoculate tissue culture flasks (100 $cm^2$ area) containing a layer of TSA. The flasks were closed and incubated at 55°C for 10 days. The subsequent spores were recovered by flooding the plate with sterilized distilled water and scraping with a spreader. The spore suspension was transferred to a centrifuge tube and the pellet resuspended in sterile water then re-centrifuged (5000g for 10 min). The spores were washed in sterile water three times and the final spore pellet resuspended in 15 ml of sterile water then heat-treated at 70°C for 15 min to inactivate vegetative cells. A dilution series was prepared in sterile water and aliquots (0.1 ml) plated onto TSA that was incubated at 55°C for up to 3 days to determine the spore density of the preparation. The spore suspension was then adjusted to a density of 7 log CFU/ml and stored at 4°C until required.

Human coronavirus 229E (HCoV-229E) was obtained from the culture collection of the Institute of Infectious Research within McMaster University (Ontario, Canada) and maintained in Opti-PRO SFM media (Thermo-Fisher). MRC-5 cells were seeded at 2 x $10^6$ cells/ml into T75 flasks in DMEM supplemented with 10% w/v FBS, 1% w/v L-glutamine, and 1% w/v pen/strep. When the cells were 90–95% confluent, the cells were washed twice with pre-warmed Opti-PRO SFM media and then infected with 5 ml of HCoV-229E in Opti-PRO SFM media at a multiplicity of infection of 0.1. Cells were incubated with virus for 80 min at 34°C with 5% $CO_2$. The cells were then washed twice with Opti-PRO SFM media and incubated with 15 ml of 3% w/v FBS DMEM containing 1% w/v sodium pyruvate, 1% w/v L-glutamine, and 1% w/v pen/strep for 5 days. On the fifth day, the supernatant was removed and clarified by centrifugation, and the virus preparation frozen at -80°C.

The $TCIS_{50}$ infectivity assay was performed by seeding Huh7.5 cells into a 96-well plate at 1.5 x $10^4$ cells/well. When the cells were 80–90% confluent they were washed twice with Opti-PRO SFM and 50 μl of a serially diluted virus was added to the cells. The cells were incubated at 34°C with 5% $CO_2$ for 5 days. On the fifth day, cell viability was assessed using Cell Titer Glo 2.0 reagent (Promega; Madison, United States). Aliquots (100 μl) Cell Titer Glo was added directly to the media, the plates were shaken for 2 minutes and then incubated for 10 min at room temperature. The luminescence was read on a Neo2 plate reader (Biotek; Winooski, United States) using luminescence fiber. The $TCID_{50}$ was calculated using log 50% endpoint dilution = log dilution showing mortality above 50%—(difference of logarithms x logarithm of dilution factor) [30].

## Gas-phase Advanced Oxidation Process reactor

The gAOP unit was provided by Clean Works Inc (Beamsville, Canada) (S1 Fig). In brief, the unit is constructed from stainless steel with the reactor housing ten UV-C lamps (23 W; 254 nm) positioned over a conveyor. The hydrogen peroxide solution was pumped to a spray head positioned at the entrance of the reactor. Ozone gas was generated by UV-184 nm lamps (12 W) positioned at either side of the unit with the gas being introduced at the conveyer level (S1 Fig). The residence time within the reactor was 30 s and controlled by the speed of the conveyor. The temperature within the reactor was maintained at 27–29°C by heated air with the UV-C, ozone and hydrogen peroxide flow-rate being monitored via sensors (S1 Fig).

## Inoculation and recovery of microbes from N95 masks

A 3 x 3 cm area was drawn on either the outer or inner surface of the N95 masks. Aliquots (0.1 ml of *ca*. 7 log CFU/ml) of the test microbe was deposited on the inner or outer surface of the mask and allowed to dry for at least 1 h prior to treatment. The masks were loaded onto

holding trays vertically or horizontally then passed through the gAOP unit that was operating with a hydrogen peroxide solution (2–6% v/v) delivered at a flow rate of 20–40 ml/min (S2 Fig). The UV-C lamps and ozone was continuously applied as the masks transitioned through the unit with a total treatment time of 30 s. The flatbed orientation was when the masks were passed through the gAOP unit twice and inverted between passes.

The inoculated masks were left for 15 min before the inoculated area was excised using a sterile blade then transferred to sterile tube containing 20 ml saline. The cells/endospores were released from the mask sections by vortexing for 60 s and a dilution series prepared in saline or sterilized water. *E. coli* was enumerated on *E. coli*/coliform Petri Films (3M, London, Canada) that were incubated at 37˚C for 24 h. For *G. stearothermphilus* enumeration, the rinse solution was heated at 70˚C for 10 min to activate the endospores. A dilution series was prepared in sterile water and plated onto TSA that was then incubated at 55˚C for up to 7 days. The remaining rinse solution and excised mask section were added to an equal volume of TSB that was subsequently incubated at 55˚C for 7 days with daily checks for growth.

HCoV-229E was inoculated onto 1 cm x 1 cm square sections drawn on masks and left to attach at room temperature for 5–30 min. The mask sections were transferred to 1 ml extraction buffer and vortexed every 5 min for 20 min at room temperature to extract the virus. The extracted virus was quantified using the $TCID_{50}$ assay pre- or post-gAOP treatment.

Toxicity testing was performed by taking 1 cm x 1 cm sections of N95 masks with one set being used directly and the other passed through the gAOP process. The mask sections were transferred to extraction buffer at a ratio of 1 ml per 1 $cm^2$ mask section then incubated for 20 min at room temperature with intermittent vortexing. Samples were then serially diluted using media and incubated with cells for 5 days before assessing viability using Cell Titer Glo 2.0.

## Fate of *E. coli* on the interior or exterior of N95 masks during inoculation and holding prior to clean flow treatment

*E. coli* K12 was cultivated in TSB overnight at 37˚C and cells harvested by centrifugation with the pellet being resuspended in saline to a final $OD_{600}$ 0.2 (8 log CFU/ml). A dilution series was prepared in saline and plated onto MacConkey agar that was incubated for 24 h at 37˚C. The cell suspension was held at 4˚C then adjusted to 7 log CFU/ml once the plate count had been determined.

Coupons (2 cm x 2 cm) were cut from 8210 N95 masks and used directly. The coupons were sub-divided into two groups. In the first group of coupons, the external surface was inoculated, while in the second group, the inner surface was inoculated. Each coupon was inoculated with 0.1 ml of inoculum and samples (N = 3) withdrawn at time 0, 15 min, 30 min, 45 min and 60 min. The individual coupons were placed in 10 ml of saline then vortexed for 30 s. A dilution series was prepared in saline then plated onto MacConkey agar that was incubated at 37˚C for 24 h.

## *Geobacillus stearothermophilus* inactivation by hydrogen peroxide, ozone or UV-C alone

Sections (3 cm x 3 cm) were excised from N95 masks and inoculated with 0.1 ml of 7 log CFU/ml *G. stearothermophilus* spore suspension. The sections (N = 3 per treatment) were dried at room temperature for 1 h before passing through the gAOP reactor with the hydrogen peroxide spray (3% v/v delivered at 40 ml/min), ozone (30 s), or UV-C (113 $mJ/cm^2$) switched on. Endospores were recovered from the treated samples and non-treated controls by suspending the sections in 20 ml saline and vortexing for 60 s. A dilution series was prepared and plated onto TSA that was subsequently incubated at 55˚C for 7 days.

## Hydrogen peroxide residue testing

Hydrogen peroxide residues on N95 masks treated with the optimized gAOP treatment were determined using MQuant Peroxide test kit (Millipore Sigma, Oakville, ON, Canada). Here, N95 masks (N = 3) were passed through the gAOP unit operating at 3% v/v hydrogen peroxide at 40 ml/min with ozone being continuously introduced and a UV-C dose of 113 mJ/cm$^2$. A 3 x 3 cm section of the N95 mask was excised and transferred to a tube containing 25 mL of distilled water then left for 1h at room temperature. The residual water was then decanted into a tube and hydrogen peroxide was determined using test strips with a low detection limit of 1.0 mg/l.

## Evaluation of filtration performance and integrity of N95 masks treated with multiple passages through the gAOP reactor

N95 masks (N = 10) that were passed through the gAOP treatment 10 or 20 times were tested for filtration and integrity by 3M Personal Safety Division Laboratory (3M, Brockville, Ontario, Canada). Filter penetration and pressure drop across the N95 masks was performed using a TSI 8130 Automated Filter Tester (AFT) set at an airflow of 85 l/min with a 200 mg NaCl challenge (Method: TEB-APR-STP-0059).

   The fit-related evaluations of each mask were performed by measuring the headband mechanical properties. Both upper and lower headbands from each N95 mask sample were evaluated for mechanical properties on an Instron model 5966 Universal Test System with a 1 kN load cell. Three elongation cycles were applied to each sample in the following order and magnitude: 200%, 50%, and 25%. These cycles simulate the donning and re-donning of a filtering facepiece respirator with non-adjustable elastic straps. The gAOP treated masks were also visually inspected for shrinkage, deformation of the nose-foam, and inner or outer shell.

## Inoculation and recovery of microbes from surgical masks

Three inoculation areas (1.5 cm diameter) were drawn on the exterior and interior of N95 masks, ensuring no overlapping between the areas (S3 Fig). Aliquots (0.1 ml) of spore suspension (7 log CFU/ml) were inoculated onto the designated areas then left for 1h at room temperature to attach. The masks were then passed through the gAOP unit and the inoculated sections were transferred to tubes containing TSB (20 ml). The spores were released by vortexing the tubes for 60 s and a dilution series prepared in saline that was subsequently plated onto TSA that was then incubated at 55°C for up to 7 days to account for the delayed germination of super-dormant endospores.

## Biological and chemical indicator for verification of gAOP mask decontamination

Apex Biological Indicator strips (MesaLabs, Bozeman, MT, USA) were applied in the current study. The stainless-steel strips are inoculated with *Geobacillus stearothermophilus* endospores (6 log CFU) on one face and have previously been applied for verification of hydrogen peroxide- based decontamination processes. Apex ribbon biological indicator strips were mounted onto N95 masks using two configurations (S4 Fig). In the first configuration, the Apex strips were attached to the outer and inner surface of the masks using double-sided tape. In Configuration 2, a window was cut from the center of the N95 masks with the Apex strip positioned with the inoculated side facing up with the other inverted (S4 Fig). In both configurations, the six masks were placed into the holder face-up and passed through the gAOP unit operating under the optimized working parameters. The Apex strips were removed from the masks after

treatment and transferred to sterile tubes into which 10 ml of TSB was added. The tubes were incubated for 48h at 55°C then checked for growth of residual survivors as per manufacture instructions.

The chemical indicator strips were fabricated by depositing 10 μl of Nafion resin 1100W (Sigma-Aldrich, ON, Canada) onto strips (1 cm x 3 cm) or disks (1 cm diameter) of thick blotting (BioRad, ON, Canada). Aliquots (10 μl) of pyrrole (Sigma-Aldrich, ON, Canada) were deposited over the Nafion and the strip/disk attached to the masks via double-sided tape then used within 10 min. The indicator was positioned within the interior and exterior of the mask with a positive reaction being recorded by a transition from pink to black.

Trials were also performed with chemical indicator strips passed through the gAOP reactor when the individual components (hydrogen peroxide, UV-C or ozone) were applied alone and then in combination. The color change of the chemical indicator was then recorded.

### Experimental design and statistics

Values represent the average of 3–6 masks per treatment with log counts of bacteria or $TCID_{50}$ values compared using one-way ANOVA in combination with the Tukey test. Statistical analysis was performed using IBM SPSS statistics software (Armok, NY, USA). Analysis of variance was used to test $P \leq 0.05$ was considered statistically significant. Fit and particulate testing represents results of 10 masks for control, ten or twenty passes through the gAOP reactor.

## Results and discussion

### Effect of gAOP operating parameters for N95 mask decontamination

The gAOP treatment was optimized in terms of mask orientation, hydrogen peroxide concentration, and flow rate with the UV-C dose and concentration of ozone remaining constant (Table 1; S2 Fig). The inactivation of *E. coli* was used as a metric to assess the different treatments given that the bacterium exhibited higher tolerance to hydroxyl-radicals compared to enveloped viruses [31]. *E. coli* was inoculated onto the exterior or interior of masks then orientated in different positions during passage through the gAOP unit operating at different hydrogen peroxide flow rates (Table 1).

With non-treated controls, the recovery of *E. coli* from the interior of the N95 masks were significantly (P<0.05) lower compared to the exterior even though the same cell density was inoculated (Table 1). To determine the underlying reasons for the lower cell density of *E. coli* recovered from the internal surfaces of N95 masks trials were performed to determine the fate

**Table 1. Log count reduction of *Escherichia coli* K12 inoculated onto the interior or exterior of N95 masks then passed through the gas phase Advanced Oxidation Process unit operating at different hydrogen peroxide flow rates.** The masks were passed through the gAOP unit flat, horizontally or vertically as depicted in S2 Fig and S1 Table.

| Treatment | Orientation | Hydrogen Peroxide Flow Rate (ml/min) | Outside Inoculated | | Inside Inoculated | |
|---|---|---|---|---|---|---|
| | | | Log CFU | Log Count Reduction | Log CFU | Log Count Reduction |
| Control Non-Treated | | | 7.60±0.12Aa | | 4.86±0.13Ab | |
| 1 | Flat bed | 40 | <1.70Ba | >5.90 | <1.70Ba | >3.16 |
| 2 | Vertical | 40 | 3.07±0.09Ca | 4.54 | <1.70Bb | >3.16 |
| 3 | Vertical | 20 | 3.51±0.05Da | 4.10 | 1.94Cb | 2.92 |
| 4 | Horizontal | 40 | <1.70Ba | >5.90 | <1.70Ba | >3.16 |
| 5 | Horizontal | 20 | 5.27±0.03Ca | 2.33 | <1.70Bb | >3.16 |

Means followed by the same upper case letter within columns are not significantly (P>0.05) different.

Means followed by the same lower case letter within rows are not significantly (P>0.05) different.

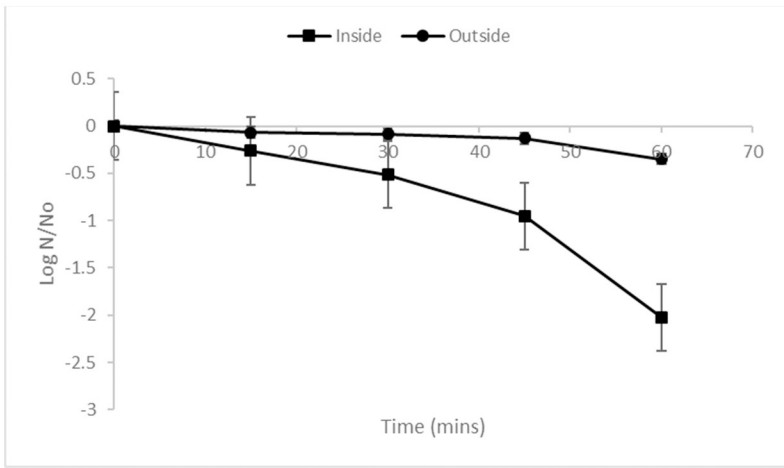

**Fig 1. Viability of *Escherichia coli* K12 on the interior or exterior of N95 masks.** *E. coli* was inoculated onto 2 cm x 2 cm sections of 3M 1820 N95 masks with samples (N = 3) taken at different times during holding at room temperature. The sections were suspended in saline and vortexed then levels of *E. coli* enumerated.

of the bacterium when inoculated inner or outer surface then held for 60 min (Fig 1; S2 Table). It was found that *E. coli* inoculated onto the outer surface of masks retained viability with no significant change in counts over the 60 min holding period. However, *E. coli* inoculated into the internal surface of N95 masks progressively declined over the 60 min holding period and resulted in a 2.02 log CFU reduction (Fig 1). It has previously been reported that the viability of *E. coli* and viruses decreases over time when introduced onto N95 masks presumably due to the electrostatic filter layers [32].

It was found that the inactivation of *E. coli* on N95 masks by gAOP treatment was independent of the hydrogen peroxide concentration (2–6% v/v) applied with no *E. coli* being recovered treated masks. It has previously been reported that a combination of hydrogen peroxide at 1–2% v/v combined with UV-C can support a 5 log CFU reduction of *Bacillus* endospores illustrating the high antimicrobial activity of hydroxyl-radicals [23]. In subsequent studies, a 3% v/v hydrogen peroxide solution was applied for the gAOP treatment given that the concentration is commercially widely available without the need for dilution.

The flow rate of hydrogen peroxide within the gAOP unit and orientation of the N95 masks during treatment affected the decontamination efficacy (Table 1). Specifically, residual *E. coli* survivors were recovered from N95 masks treated with gAOP operating at 20 ml/min (Table 1). However, no *E. coli* was recovered from N95 masks treated with gAOP when the hydrogen peroxide was introduced at 40 ml/min (Table 1). The results indicate that the efficacy of the gAOP was dependent on the degree of misting by hydrogen peroxide to cover the N95 masks and to generate sufficient hydroxyl radicals to inactivate *E. coli*. The importance of misting was also highlighted by the effect on the orientation of the mask. N95 masks passed through the gAOP unit horizontally (Treatment 4 and 5; Table 1) resulted in a higher decontamination efficacy compared with those positioned vertically (Treatment 2 and 3; Table 1). No *E. coli* were recovered from masks that had been double passed (flatbed) through the gAOP being inverted after the first pass (Treatment 1; Table 1). The effect of position again reflects the exposure to hydroxyl radicals to the surfaces on the outside and the internal surface of masks.

The orientation of masks is critical in UVGI treatments given that the surfaces are required to be directly exposed to the UV-C photons [12]. With hydrogen peroxide vapor the orientation is less critical but there is a need to facilitate free-flow of the antimicrobial gas around the masks [6]. In a similar manner, gAOP treatment requires the circulation of hydroxyl-radicals to the surfaces of the mask but do not need to be directly exposed to UV as with gas plasma treatment [33].

### Decontamination of N95 masks inoculated with *Geobacillus stearothermophilus* endospores and human coronavirus and treated using the gAOP reactor

Unlike *E. coli*, the levels of endospores recovered from the internal surface of the N95 masks were not significantly different (P>0.05) compared to the inner layers (Table 2). No surviving endospores were recovered from N95 masks passed through the gAOP unit thereby verifying the antimicrobial efficacy of the process (Table 2). The results are in agreement with studies performed on the sporicidal activity of hydroxyl-radical activity in carton sterilization [25]. To verify that endospore inactivation was via hydroxyl-radicals trials were performed to assess the lethality of the individual components of the gAOP process. Here, mask sections were inoculated with *Geobacillus* endospores were passed through the reactor with only the hydrogen peroxide spray (3% v/v at 40 ml/min), ozone or UV-C lamps (dose 113 mJ/cm$^2$) turned on. The log count reductions obtained were 0.97±0.07 log CFU for hydrogen peroxide, 0.10±0 log CFU for ozone and 0.93±0.09 log CFU for UV-C alone. The results confirm that the generation of hydroxyl-radicals exhibited a synergistic lethality compared to the individual component parts.

The initial TCID$_{50}$ of CoV-229E inoculated onto N95 masks was determined to be >2 x 10$^8$ with no survivors being detected on samples treated with gAOP (S3 Table). The results confirm the susceptibility of coronavirus to hydroxyl-radicals [31]. There was no significant difference (*P>0.05*) in the viability of Huh7.5 cells exposed to the extract taken from mask sections that had been treated with gAOP (0.939± 0.017 RLU) relative to non-treated controls (0.867 ±0.012 RLU) (S4 Table). The result verifies that no toxic products were generated as a result of the gAOP process.

### Filtration performance testing of gAOP treated N95 masks

The penetration test illustrated that N95 masks treated with the gAOP process were not significantly different (*P>0.05*) from non-treated controls and within the 0.1–2.0% tolerance limits (Table 3). The insignificant difference in the pressure drop across gAOP treated masks compared to controls also provided evidence that the N95 masks retained functionality.

The headband properties of the gAOP treated N95 masks were not significantly (*P>0.05*) differ compared to the controls and no visual deformation, shrinkage, or change in texture was found. Collectively the results confirmed the N95 masks treated up to 20 cycles in the gAOP unit did not alter the performance or fit of masks.

### Inactivation of *Geobacillus stearothermophilus* endospores inoculated on the interior and exterior of surgical masks then treated using the gAOP process

Surgical masks were inoculated at different positions on the interior and exterior of surgical masks then passed through the gAOP reactor operating under optimized conditions (Table 4;

**Table 2. Inactivation of *Geobacillus stearothermophilus* endospores inoculated onto the internal and external surfaces of 8210 N95 masks then passed through the gAOP process on a fully loaded tray (S5 Fig).**

| | | | Samples | Log CFU/ml |
|---|---|---|---|---|
| | | **N95 Mask** | Inoculum | 7.02±0.03 |
| | | | Control | Log CFU/mask |
| | | | External | 6.08±0.12 |
| | | | Internal | 6.04±0.16 |
| | **Inside/Outside Mask** | **Inoculation Area** | **Log CFU/section (Negative/Positive by Enrichment)** | **Log Reduction** |
| **Holder Position 1** | Outside | A | Negative | 6.08 |
| | | B | Negative | 6.08 |
| | | C | Negative | 6.08 |
| | Inside | D | Negative | 6.04 |
| | | E | Negative | 6.04 |
| | | F | Negative | 6.04 |
| **Holder Position 2** | Outside | A | Negative | 6.08 |
| | | B | Negative | 6.08 |
| | | C | Negative | 6.08 |
| | Inside | D | Negative | 6.04 |
| | | E | Negative | 6.04 |
| | | F | Negative | 6.04 |
| **Holder Position 3** | Outside | A | Negative | 6.08 |
| | | B | Negative | 6.08 |
| | | C | Negative | 6.08 |
| | Inside | D | Negative | 6.04 |
| | | E | Negative | 6.04 |
| | | F | Negative | 6.04 |
| **Holder Position 4** | Outside | A | Negative | 6.08 |
| | | B | Negative | 6.08 |
| | | C | Negative | 6.08 |
| | Inside | D | Negative | 6.04 |
| | | E | Negative | 6.04 |
| | | F | Negative | 6.04 |
| **Holder Position 5** | Outside | A | Negative | 6.08 |
| | | B | Negative | 6.08 |
| | | C | Negative | 6.08 |
| | Inside | D | Negative | 6.04 |
| | | E | Negative | 6.04 |
| | | F | Negative | 6.04 |
| **Holder Position 6** | Outside | A | Negative | 6.08 |
| | | B | Negative | 6.08 |
| | | C | Negative | 6.08 |
| | Inside | D | Negative | 6.04 |
| | | E | Negative | 6.04 |
| | | F | Negative | 6.04 |

Inoculation sites (A, B, C) are the exterior surface of the N95 mask with D, E and F being on the internal surface (S5 Fig).

**Table 3. Evaluation of filter penetration, pressure drop and headband integrity of N95 masks passed through the gas phase-Advanced Oxidation Process decontamination treatment 10 or 20 times compared to non-treated controls (S5 and S6 Tables).**

| Transitions through the gAOP unit | Filter Penetration Test (%) | Pressure Drop (mm Hg) | Headband Force 3rd Cycle 50% Upper/Lower |
|---|---|---|---|
| No Treatment | 0.15±0.05A | 7.53±0.29A | 0.38±0.01/0.38±0.01 |
| 10 | 0.17±0.09A | 7.57±0.36A | Not Determined |
| 20 | 0.17±0.05A | 7.68±0.22A | 0.37±0.01/0.37±0.01 |

Means followed by the same letter within columns are not significantly different (P>0.05).

**Table 4. *Geobacillus* endospore loading inoculated onto surgical masks and passed through the gas phase Advanced Oxidation Process reactor.** Non-treated controls were used to determine the initial endospore levels.

| Outside | Log CFU (Enrichment) | LCR | Inside | Log CFU (Enrichment) | LCR |
|---|---|---|---|---|---|
| Control | | | | | |
| Mask I Outside | 6.44±0.12 | | Mask I Inside | 6.16±0.08 | |
| Mask II Outside | 6.58±0.34 | | Mask II Inside | 6.35±0.11 | |
| Mask III Outside | 6.41±0.29 | | Mask III Inside | 6.31±0.21 | |
| Average | 6.48±0.09 | | Average | 6.27±0.07 | |
| Clean Flow | | | | | |
| Mask 1 | | | Mask 1 | | |
| A* | Negative | 6.48 | D | Negative | 6.27 |
| B | Negative | 6.48 | E | Negative | 6.27 |
| C | Negative | 6.48 | F | Negative | 6.27 |
| Mask 2 | | | Mask 2 | | |
| A | Negative | 6.48 | D | Negative | 6.27 |
| B | Negative | 6.48 | E | Negative | 6.27 |
| C | Negative | 6.48 | F | Negative | 6.27 |
| Mask 3 | | | Mask 3 | | |
| A | Negative | 6.48 | D | Negative | 6.27 |
| B | Negative | 6.48 | E | Negative | 6.27 |
| C | Negative | 6.48 | F | Negative | 6.27 |
| Mask 4 | | | Mask 4 | | |
| A | Negative | 6.48 | D | Negative | 6.27 |
| B | Negative | 6.48 | E | Negative | 6.27 |
| C | Negative | 6.48 | F | Negative | 6.27 |
| Mask 5 | | | Mask 5 | | |
| A | Negative | 6.48 | D | Negative | 6.27 |
| B | Negative | 6.48 | E | Negative | 6.27 |
| C | Negative | 6.48 | F | Negative | 6.27 |
| Mask 6 | | | Mask 6 | | |
| A | Negative | 6.48 | D | Negative | 6.27 |
| B | Negative | 6.48 | E | Negative | 6.27 |
| C | Negative | 6.48 | F | Negative | 6.27 |

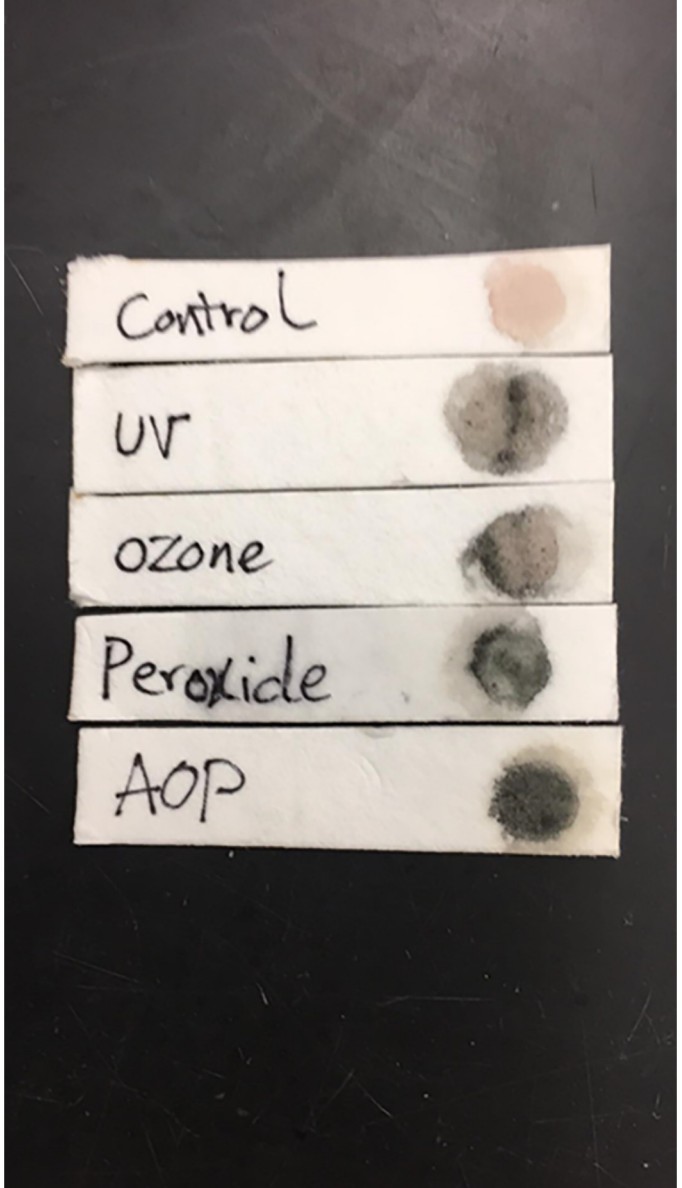

**Fig 2. Color change in chemical indicator based on pyrrole-Nafion films.** Nafion was deposited onto blotting paper strips and overlaid with pyrrole. The strips were then passed through the Gas Phase Advanced Oxidation Process reactor operating at 3% v/v hydrogen peroxide delivered at 40 ml/min, UV-C dose of 113 mJ/cm2 and ozone being delivered through the 30 s treatment. Additional strops were passed through the gAOP unit with hydrogen peroxide, UV-C or ozone alone. A non-treated control acted as a comparison.

S3 Fig). No spores were recovered from the treated surgical masks verifying that the gAOP treatment could successfully decontaminate both the interior and exterior surfaces.

## Biological and chemical indicator for verifying decontamination performance of gAOP treatment

The biological indicator strips were affixed to the internal or external surface of N95 masks or bridging a cut-out section (S4 Fig). The masks were then passed through the gAOP reactor

operating under the developed working parameters (3% v/v hydrogen peroxide delivered at 40 ml/min with a UV-C dose of 113 mJ/cm$^2$ with ozone being applied throughout the 30 s process). None of the Apex biological indicator strips tested negative when passed through the gAOP reactor irrespective of being position in Configuration 1 or 2 (S4 Fig). In contrast, the non-treated Apex biological strips tested positive thereby confirming that the *Geobacillus* endospores were viable. The results confirm that the Biological Indicator *Geobacillus* endospore strips could be applied for routine verification of the gAOP process.

The chemical indicator serves to differentiate between those N95 masks that have been treated using gAOP from those awaiting treatment. Besides, indicators also serve to verify that the treatment has been adequately applied so can be viewed as a complement to biological indicators. In the current study, the Nafion-Pyrrole indicator transitioned from pink to blue-black when passed through the gAOP reactor operating under the aforementioned working parameters. However, when each of the inputs to the gAOP process (i.e UV-C, hydrogen peroxide, or ozone) were applied alone the color change was less distinct and incomplete (Fig 2). The results are in agreement with other reports demonstrating that the generation of hydroxyl-radicals can support the oxidative polymerization of pyrrole [29,34].

## Conclusions

The study was directed towards validating and verifying the efficacy of a gAOP to decontaminate N95 and surgical masks without resulting in detrimental changes in functionality. In addition, the utility of a biological and chemical indicator to verify decontamination performance was also performed. It was found that the gAOP treatment could support the inactivation of human coronavirus, vegetative bacterial cells, and endospores without causing negative effects on mask functionality. Given the continuous, rapid, nature of the process, it could be envisaged that the gAOP unit could be applied on-site (for example, hospital ward) for recycling or extending the use of masks. The different operating parameters (hydrogen peroxide, ozone, and UV-C dose) can be continuously monitored in real-time thereby resulting in a controlled process. Yet, a biological and chemical indicator has been identified in the current study that can be applied for process verification. The current study tested a limited number of masks due to availability issues and future studies will be looking at a broader range of N95 mask types.

## Supporting information

**S1 Fig. Gas-phase Advanced Oxidation Process unit for N95 and surgical mask decontamination.** The masks were loaded onto holders then passed through a hydrogen peroxide mist with ozone gas being introduced via side vents. The masks then pass under UV-C lamps with a total transit time of 30s. The flow rate of hydrogen peroxide, ozone concentration and UV-C intensity monitored via sensors.
(DOCX)

**S2 Fig. Orientation of N95 masks inoculated with *Escherichia coli* K12 during passage through the gas phase Advanced Oxidation Process unit.** Treatment 1 was when the masks were placed down for the first pass then inverted for the second pass. Treatment 2 and 3 was when the N95 masks were held vertical and Treatment 4 and 5 when held horizontally during passage through the reactor.
(DOCX)

**S3 Fig. Surgical masks inoculated with *Geobacillus stearothermophilus* endospores prior to passing through the gas phase Advanced Oxidation Process reactor.** The *Geobacillus* endospores (0.1 ml of 7 log CFU/ml) was inoculated into the marked areas on the outside (A, B, C)

or inside (D, E, F) of surgical masks then allowed to dry for at least one hour before treated with the gAOP reactor.
(DOCX)

**S4 Fig. 1 Biological (*Geobacillus stearothermophilus*) indicator strips positioned on the exterior or interior of N95 masks (Configuration 1) or bridged across a cut out section (Configuration 2).** In each configuration the N95 masks were run through the Clean Flow unit positioned face-up.
(DOCX)

**S5 Fig. N95 masks inoculated onto the marked areas on the interior (A, B, C) or exterior (D, E, F) with *Geobacillus* spores (0.1 ml of 7 log CFU/ml) then passed through the gas phase Advanced Oxidation Process reactor.**
(DOCX)

**S1 Table. Log count reduction of *Escherichia coli* K12 inoculated onto the interior or exterior of N95 masks then passed through the gas phase Advanced Oxidation Process unit operating at different hydrogen peroxide flow rates.** The masks were passed through the gAOP unit flat, horizontally or vertically as depicted in S2 Fig.
(DOCX)

**S2 Table. Viability of *Escherichia coli* K12 on the interior or exterior of N95 masks.** *E. coli* was inoculated onto 2 cm x 2 cm sections of 3M 1820 N95 masks with samples (N = 3) taken at different times during holding at room temperature. The sections were suspended in saline and vortexed then levels of *E. coli* enumerated.
(DOCX)

**S3 Table. $TCID_{50}$/ml of human coronavirus E299 inoculated onto mask sections then treated with gas phase Advanced Oxidation Process.**
(DOCX)

**S4 Table. Normalized cell viability before and after gas phase Advanced Oxidation Process.**
(DOCX)

**S5 Table. Evaluation of filter penetration and pressure drop of N95 masks passed through the gas phase-Advanced Oxidation Process decontamination treatment 10 or 20 times compared to non-treated controls.**
(DOCX)

**S6 Table. Evaluation of headband integrity of N95 masks passed through the gas phase-Advanced Oxidation Process decontamination treatment 10 or 20 times compared to non-treated controls.**
(DOCX)

## Acknowledgments

We thank Clean Works Inc for providing access to a Clean Flow unit and 3M for undertaking the mask evaluation testing.

## Author Contributions

**Conceptualization:** Mahdiyeh Hasani, Keith Warriner.

**Data curation:** Tracey Campbell, Fan Wu.

**Formal analysis:** Mahdiyeh Hasani, Tracey Campbell, Keith Warriner.

**Investigation:** Mahdiyeh Hasani.

**Supervision:** Keith Warriner.

**Writing – original draft:** Mahdiyeh Hasani, Tracey Campbell, Fan Wu, Keith Warriner.

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
