## [Decision Letter · Decision Letter 0]

8 Feb 2021

PONE-D-21-00533

Decontamination of N95 and surgical masks using a treatment based on a continuous gas phase-Advanced Oxidation Process

PLOS ONE

Dear Dr. Warriner,

Thank you for submitting your manuscript to PLOS ONE. After careful consideration, we feel that it has merit but does not fully meet PLOS ONE’s publication criteria as it currently stands. Therefore, we invite you to submit a revised version of the manuscript that addresses the points raised during the review process.

We look forward to receiving your revised manuscript.

Kind regards,

Rashid Nazir

Academic Editor

PLOS ONE

Journal Requirements:

Reviewers' comments:

Reviewer's Responses to Questions

**Comments to the Author**

1. Is the manuscript technically sound, and do the data support the conclusions?

Reviewer #1: Yes

Reviewer #2: Yes

2. Has the statistical analysis been performed appropriately and rigorously? 

Reviewer #1: No

Reviewer #2: Yes

3. Have the authors made all data underlying the findings in their manuscript fully available?

Reviewer #1: No

Reviewer #2: Yes

4. Is the manuscript presented in an intelligible fashion and written in standard English?

Reviewer #1: No

Reviewer #2: Yes

5. Review Comments to the Author

Reviewer #1: PLOS ONE

Manuscript: PONE-D-21-00533

Article Title: Decontamination of N95 and surgical masks using a treatment based on a continuous gas phase-Advanced Oxidation Process

Comments and recommendations:

1. Authors have proposed a very effective gas-phase Advanced Oxidation Process-based decontamination approach for decontaminating surgical masks N95 and 455 has good evidence of inactivation of human coronavirus and bacterial cells.

2. I went through entire manuscript and found that the paper is logically arranged and reflect the coverage of main thematic area chosen for the study, however some Major improvements are recommended. English language and sentence structures are poorly written, major improvements needed; some minor grammatical errors need to be adjusted throughout the manuscript. Rearranging all the manuscript sections with proper heading and rephrasing of the sentences throughout the manuscript is recommended.

3. Authors are supposed to provide pathways explaining the reason for the role of hydrogen peroxide combined with UV-C lamps and ozone sources. Please provide the evidence how lethality of gAOP was independent of the applied hydrogen peroxide concentration. Since hydrogen peroxide is an antiseptic and bleaching agent that is common and well-known. When exposed to moisture and bubbled with ozone, atmospheric gas and toxic smog components, a strong and extremely microbicidal chemical reaction is generated that can kill almost any form of bacteria and make them harmless.

4. There is no fair justification for research goals and related findings.

5. Clearly mention concentrations of the reagents (Ozone, hydrogen peroxide) and the light intensity with spectrum.

6. Author surveyed the literature and supported the facts with available literature, but it is recommended to add more supportive literature in introduction as well as results and discussion section, repeating one reference in every section is not recommended. Literature reviewed is not justified to focus on research parameters.

7. Consistency among the paragraphs is essential in any article, otherwise reader shall not be able to understand what is written. It is recommended for the authors to relook the sentences and paragraphs and provided consistency among the words and paragraphs especially in the introduction section.

8. The conclusion is well written with proper future direction along with the facilities required. All the objectives must be properly stated with concluding results.

9. Concluding Remarks:

Keeping in view the observations in the comments section, this research paper needs major revisions. Some minor and moderate revision are also suggested for rephrasing and grammatical error explained in comments section.

The reviewer would like to recommend this article after assurance of the incorporation of all the observations and comments for the possible publication in PLOS ONE.

Dr Nadia Riaz, PhD

Assistant Professor

Department of Environmental Science,

COMSATS University Islamabad, Abbottabad Campus,

Abbottabad, Pakistan

nadiariazz@gmail.com/ nadiariazz@cuiatd.edu.pk

Reviewer #2: Major Concern

Is there any negative effect of pyrrole on the filtration efficacy of mask? Is there any allergic response associated with pyrrole?

Tables and figures should be self-explanatory. Please explain abbreviations used in captions and legends.

References are not properly formatted and some are even incomplete.

I shall suggest to improve quality of figures. Some of the figures showing equipment and sample loading might be shifted to supplementary data.

Minor revisions

Line 77; It would be appropriate to write ‘spp.’ after genus.

Line 88-89; typo error, ‘.’ Should be replaced by ‘,’

Line 100; would it be not appropriate to replace ‘fresh produce’ with some more obvious phrase?

Line 141; the word slope should be replaced by ‘slant’

Line 142: The word sterile should be replaced by ‘sterilized’

Line 149-150; the purpose of the activity performed is not described.

Line 184-188; Why figure caption is provided here without figure?

Line 191; Cell density should also be mentioned here.

Line 210: K12 should not be italicised.

Line 260 and 277; why there are different incubation times? Please explain.

6. PLOS authors have the option to publish the peer review history of their article (what does this mean?). If published, this will include your full peer review and any attached files.

Reviewer #1: No

Reviewer #2: **Yes: **Dr. Muhammad Ali, Department of Biotechnology, COMSATS University Islamabad, Abbottabad campus, Pakistan

---

## [Author Response · Author response to Decision Letter 0]

12 Feb 2021

Response to Reviewers 

We thank the reviewers to take time for reviewing the script and raising constructive comments that we have addressed as indicated below. 

Reviewer #1 

English language and sentence structures are poorly written, major improvements needed; some minor grammatical errors need to be adjusted throughout the manuscript. Rearranging all the manuscript sections with proper heading and rephrasing of the sentences throughout the manuscript is recommended. The script has been further proof-read. 

Authors are supposed to provide pathways explaining the reason for the role of hydrogen peroxide combined with UV-C lamps and ozone sources. Please provide the evidence how lethality of gAOP was independent of the applied hydrogen peroxide concentration. Since hydrogen peroxide is an antiseptic and bleaching agent that is common and well-known. When exposed to moisture and bubbled with ozone, atmospheric gas and toxic smog components, a strong and extremely microbicidal chemical reaction is generated that can kill almost any form of bacteria and make them harmless. By using hydrogen peroxide, ozone or UV-C alone there were negligible reduction of endospores. Specifically, the 3% hydrogen peroxide was 0.97�0.07 log cfu reduction, ozone (30s) 0.10�0 log cfu and 0.93 � 0.09 log reduction for UV-C alone. Endospores are relatively resistant to low concentrations of hydrogen peroxide with 35% required to achieve a 6 log reduction. 

There is no fair justification for research goals and related findings. The need for a decontamination method for masks originated from the request from health agencies to address the on-going shortage of PPE. This has been further underlined in the revised script. 

Clearly mention concentrations of the reagents (Ozone, hydrogen peroxide) and the light intensity with spectrum. The reaction conditions are provided within the abstract and throughout the text. 

Author surveyed the literature and supported the facts with available literature, but it is recommended to add more supportive literature in introduction as well as results and discussion section, repeating one reference in every section is not recommended. Literature reviewed is not justified to focus on research parameters. There are relatively few published works on N95 mask decontamination and application of gas phase AOP. Consequently, several of the references are repeated through the Introduction and Discussion. I 

Consistency among the paragraphs is essential in any article, otherwise reader shall not be able to understand what is written. It is recommended for the authors to relook the sentences and paragraphs and provided consistency among the words and paragraphs especially in the introduction section. The script has been further proof-read to ensure consistency. 

The conclusion is well written with proper future direction along with the facilities required. All the objectives must be properly stated with concluding results. The conclusion has been revised to include the objectives of the study. 

Reviewer #2: 

Is there any negative effect of pyrrole on the filtration efficacy of mask? Is there any allergic response associated with pyrrole? The actual indicator would be adhered to the mask using double-sided tape and hence no direct contact between Nafion-pyrrole with the filter material. One could suspect the chemical indicator would be placed away from the filter material (for example, strap) or removed prior to donning the mask. 

Tables and figures should be self-explanatory. Please explain abbreviations used in captions and legends.

References are not properly formatted and some are even incomplete. The legends have been corrected in the revised script. 

I shall suggest to improve quality of figures. Some of the figures showing equipment and sample loading might be shifted to supplementary data. The suggested figures have been transferred to supplemental data. 

Minor revisions

Line 77; It would be appropriate to write ‘spp.’ after genus. Corrected in the revised script.

Line 88-89; typo error, ‘.’ Should be replaced by ‘,’ Corrected in the revised script.

Line 100; would it be not appropriate to replace ‘fresh produce’ with some more obvious phrase? The term “fresh produce” has been replaced with “fresh fruit and vegetables”. 

Line 141; the word slope should be replaced by ‘slant’ Slant has been included in the revised script. 

Line 142: The word sterile should be replaced by ‘sterilized’ Sterilized has been included in the revised script.

Line 149-150; the purpose of the activity performed is not described. This was to determine the spore density of the suspension and was included in the revised script. 

Line 184-188; Why figure caption is provided here without figure? This was to illustrate where the figure would be included. In this case the figure has been moved to supplemental data. 

Line 191; Cell density should also be mentioned here. The inoculating suspension was ca 7 log cfu/ml. This has been included within the revised script. 

Line 210: K12 should not be italicised. Corrected in the revised script. 

Line 260 and 277; why there are different incubation times? Please explain. In sterility testing it is common practice to have a prolonged incubation to account for super-dormant endospores that exhibit delayed germination. In the current study there was no growth observed in the tubes incubated over 7-days thereby confirming spore inactivation. This has been clarified in the revised script.

---

## [Editor Report · Decision Letter 1]

1 Mar 2021

Decontamination of N95 and surgical masks using a treatment based on a continuous gas phase-Advanced Oxidation Process

PONE-D-21-00533R1

Dear Dr. Warriner,

We’re pleased to inform you that your manuscript has been judged scientifically suitable for publication and will be formally accepted for publication once it meets all outstanding technical requirements.

Kind regards,

Rashid Nazir

Academic Editor

PLOS ONE
---

## [Editor Report · Acceptance letter]

4 Mar 2021

PONE-D-21-00533R1 

Decontamination of N95 and surgical masks using a treatment based on a continuous gas phase-Advanced Oxidation Process 

Dear Dr. Warriner:

I'm pleased to inform you that your manuscript has been deemed suitable for publication in PLOS ONE. Congratulations! Your manuscript is now with our production department. 

Kind regards, 

on behalf of

Dr Rashid Nazir 

Academic Editor

PLOS ONE